# *Acanthamoeba* Keratitis, Pathology, Diagnosis and Treatment

**DOI:** 10.3390/pathogens10030323

**Published:** 2021-03-10

**Authors:** Nicholas Fanselow, Nadia Sirajuddin, Xiao-Tang Yin, Andrew J. W. Huang, Patrick M. Stuart

**Affiliations:** 1Department of Ophthalmology, Saint Louis University, Saint Louis, MO 63104, USA; nicholas.fanselow@health.slu.edu (N.F.); nadia.sirajuddin@health.slu.edu (N.S.); xiaotang.yin@health.slu.edu (X.-T.Y.); 2Department of Ophthalmology and Visual Sciences, Washington University, Saint Louis, MO 63110, USA; huangandrew@wustl.edu

**Keywords:** *Acanthamoeba*, keratitis, pathogenesis, diagnosis, therapy

## Abstract

*Acanthamoeba* keratitis is an unusual corneal infection that is recently increasing in frequency and is often contracted by contact lens wearers, someone who experienced recent eye trauma, or someone exposed to contaminated waters. *Acanthamoeba* survive in air, soil, dust, and water. Therefore, eye trauma and poor contact lens hygiene practices lead to the entrapment of debris and thus infection. *Acanthamoeba* keratitis results in severe eye pain, inflammation, and defects of the epithelium and stroma that can potentially result in vision loss if not diagnosed early and treated promptly. The disease can be diagnosed using corneal scrape/biopsy, polymerase chain reactions, impression cytology, or in vivo confocal microscopy. Once diagnosed, it is usually treated with an antimicrobial combination therapy of biguanide and aromatic diadine eye drops for several months. Advanced stages of the disease result in vision loss and the need for corneal transplants. Avoiding the risk factors and diagnosing the disease early are the most effective ways to combat *Acanthamoeba* keratitis.

## 1. Introduction

*Acanthamoeba* is a common free-living amoeba that is found in many different environmental niches and can be isolated from water bodies, drainages, sediments, surgical instruments, dialysis units, skin lesions, and to the point of this review, contact lenses [1]. These organisms are considered an emerging parasite primarily due to the difficulty of treating infections by these organisms [2]. Interestingly, these organisms can inhabit both in external environments as well as within the bodies of hosts they infect [3].

*Acanthamoeba* has the capacity to cause severe infections among immunocompetent as well as immunocompromised individuals [4]. The infections caused by *Acanthamoeba* include *Acanthamoeba* keratitis (AK) among healthy individuals, especially those wearing contact lenses, and a life threatening Granulomatous amoebic encephalitis (GAE) infection of the central nervous system (CNS) among immunocompromised population [5,6]. Although the incidence of AK/GAE is different around the world (0.13–33 cases per million), the rates have been steadily increasing throughout recent years, especially in developed countries [7,8,9,10,11,12,13]. Since GAE is found in immunocompromised individuals, the mortality rate is quite high ranging from 95 to 98% among infected patients [14,15]. This high mortality rate is due to several factors, including late diagnosis or misdiagnosis of the infection and/or lack of effective therapeutic agents against the resistant cyst forms of *Acanthamoeba*. 

*Acanthamoeba* keratitis is a rare infection of the cornea that, if not diagnosed and treated properly, can be a sight threatening disease. While *Acanthamoeba* keratitis (AK) is not as common as other corneal infections, such as bacterial or viral keratitis, it presents itself with its own set of challenges, particularly in terms of early diagnosis and appropriate treatment. *Acanthamoeba* are currently divided into twenty-two different genotypes (T1–T22) based on 18s rRNA gene typing [16]. Genotyping of *Acanthamoeba* is important because the different genotypes show variation in clinical presentation and response to medical therapy [17]. At least eight of the genotypic classes (T2, T3, T4, T5, T6, T10, T11, and T15) have been shown to cause AK, with the most common causative genotype being T4 [17,18]. The most common infectious species are *Acanthamoeba castellani* and *Acanthamoeba polyphaga*, both from the T4 genotype [16,19]. The pathogen is transmitted through corneal contact with a contaminated substance. Most cases seen in humans seem to be associated with contaminated water, soil, or trauma to the eye [20]. Once in the eye, the amoebae feed on the keratocytes with invasion and destruction of corneal stroma. Since AK is a relatively uncommon corneal infection, it often goes undiagnosed and untreated for long periods of time. This leads to a delayed institution of appropriate treatments with a greater ratio of unfavorable visual outcomes.

## 2. Pathogenesis

The progression of *Acanthamoeba* keratitis occurs in two main phases. An initial phase where infiltration is limited to the corneal epithelium, and a secondary phase where the parasite invades the underlying stroma. Once in the stroma, extensive damage to the collagen matrix occurs which provokes intense inflammation [21]. Treatment during the initial stages of pathogenesis is more successful than treatment during later stages of disease, which is why early diagnosis and treatment are essential. 

The first step in the pathogenesis of AK is the adhesion of the microbe to the corneal surface. The process of adhesion is mediated by a number of proteins, the most important of which has been identified as a mannose-binding protein expressed by the amoeba [21,22]. The process continues with *Acanthamoeba* trophozoites breaking down the epithelial barrier by mechanisms of direct cytolysis, phagocytosis, and induction of apoptosis [22]. Following adhesion and breakdown of the corneal epithelium, trophozoites invade the underlying collagenous stroma. The process of stromal invasion is mediated by a number of products of the amoeba, including metalloproteinases and serine proteinases. These proteinases work to produce a potent cytopathic effect that kills host cells and degrades the epithelial basement membrane as well as the stromal matrix to progress into deeper layers of the cornea [21,22]. Stromal involvement is typically seen late in the course of AK. Once in the stroma, the trophozoites feed on keratocytes and organic particles causing keratocyte depletion, induction of an intense inflammatory response, and finally stromal necrosis [21,22]. These latest stages of disease have potentially sight-threatening effects [23].

*Acanthamoeba* also have two stages to their life cycle, a vegetative trophozoite stage and a dormant cystic stage. *Acanthamoeba* trophozoites can undergo encystment into the dormant form through the influence of host resting macrophages [22]. These *Acanthamoeba* cysts pose a serious risk of recurrence in patients who have been previously treated for AK. These two forms of the amoeba have important implications for management of the disease.

## 3. Risk Factors

The most common mechanism of contracting *Acanthamoeba* keratitis is corneal trauma or direct exposure to contaminated waters such as using non-sterile lens solution, swimming in unchlorinated pool and lakes. While these are the most common mechanisms for contracting AK, the greatest risk factor associated with AK is wearing contact lenses. Contact lenses can trap pathogens on the cornea or cause corneal abrasions and provide an environment for the organisms to thrive. It is reported that up to 85% of all AK cases are in contact lens wearers [24]. Typical issues seen in those wearing contact lenses that increase the risk of AK are over-night contact lens wearing or extended storage of contacts in non-sanitized solutions such as tap water or unpreserved saline solution [25,26]. In both situations, the amoeba can thrive in the favorable environment for a substantial amount of time and cause infection when contacts are re-inserted onto the cornea. Appropriate contact lens hygiene can prevent unexpected infection. However, most people do not adhere to contact lens hygiene properly. For this reason, multipurpose contact lens solutions should be used when storing and washing contacts. Hydrogen-peroxide based systems have the greatest efficacy toward killing *Acanthamoeba* and are significantly more effective than non-sterile saline solutions or chlorine-based solutions [27]. Efficacy of hydrogen-peroxide based systems may still be limited, as they are not always effective in killing all *Acanthamoeba* cysts [24]. AK is less frequently encountered by those who do not wear contact lenses. If the infection presents itself in a person who does not wear contact lenses, it is often due to minor corneal trauma of the eye associated with infectious debris abrading the cornea. This is the common mechanism seen in developing countries where the prevalence of wearing contacts is not as high [19]. The prevalence of AK is also higher in men than women, most likely due to a less strict hygiene regimen or more outdoor activities. AK is more often seen among younger adults, once again most likely due to hygiene habits or more outdoor exposures. Nonetheless, AK is also more seen among those older than 53, likely associated with corneal changes due to aging [20].

## 4. Signs/Symptoms

Diagnosing AK based on clinical presentation can be difficult because initial signs and symptoms resemble other corneal conditions. Similar to other corneal infections, the initial symptoms of AK are relatively nonspecific. The affected patient may only have minor ocular irritations, tearing or blurred vision. However, severe ocular or periocular pain is often the hallmark of AK associated with the progression of the infection and intense stromal inflammation [28,29]. 

The signs of *Acanthamoeba* keratitis are unilateral most of the time and progress slowly, beginning superficially at the epithelium and eventually affecting the stroma. Clinical findings of this conditions are illustrated in Figure 1 and Figure 2. Figure 1A,B is from a 40 year-old male contact user with unilateral, coarse superficial punctate epithelial erosions. Figure 2 is from a 74 year-old woman contact lens wearer with a dense ring stromal infiltrate. These figures reiterate the fact that as already described, this disease typically takes place in contact lens wearers. When one considers what typically occurs during this disease, the first signs begin to appear as a diffuse superficial keratopathy [26] as illustrated in Figure 1. Within the first two weeks of infection, the eye can undergo chameleon-like epithelial changes referred to as “dirty epithelium” [30]. These changes include a pseudo-dendritiformic epitheliopathy with grey epithelial opacities [30]. This dendritic keratitis can be misdiagnosed as viral keratitis caused by herpes simplex or herpes zoster. However, pseudodendrites caused by AK are characteristically different because its epithelium defects have no involvement of the endothelium and lack the widening terminal knots (known as epithelial dendrites), both of which are commonly seen in herpes sim-plex keratitis [26,31]. Furthermore, the patient with herpetic keratitis may experience significantly decreased corneal sensation [32]. Unusual infiltrates along radiating corneal nerves (known as radial keratoneuritis) may present as the disease progresses, with a paracentral ring infiltrate such as seen in Figure 2 being a pathognomonic sign of AK. However, multiple studies have reported a significant number of AK cases that do not exhibit the classic ring infiltrates [26,33]. Ring infiltrates could be seen among corneal ulcers and fungal keratitis. It is noteworthy, unlike fungal keratitis, AK are more frequently associated with epithelium defects and perineural stromal infiltrates [26]. In addition, stromal infiltrates caused by AK are often multifocal rather than monofocal as seen typically in bacterial keratitis [26]. 

## 5. Diagnosis

When it comes to *Acanthamoeba* keratitis, high index of clinical suspicion and early diagnosis are essential to avoid untoward health outcomes for patients. The first step to diagnosing AK is having a clinical suspicion of the disease. Since AK is uncommon compared to the other causes of keratitis, it is often overlooked as a differential diagnosis. AK should be taken into consideration in anyone exhibiting the risk factors as described above, especially contact lenses wearers, or anyone who is demonstrating severe ocular pain. To complicate the matters, there are also many reports of AK in mixed form with viral, bacterial, or fungal pathogens also present [34,35]. These forms of mixed keratitis have important implications for diagnosis and management of disease. Early confirmation of the infectious agents leads to swift and effective treatment with earlier recovery before occurrence of any serious damage to the cornea and vision. There are multiple options available that can assist in the diagnosis of AK, and often multiple techniques are used to ensure proper diagnosis.

### 5.1. Corneal Scraping

Performing a corneal scraping for microbial culture to identify the causative pathogen(s) is generally considered as the gold standard clinically to confirm AK. Although culture tests carry a high specificity (100%), they generally lack strength regarding sensitivity, which ranges from 7–66.7% depending upon culture techniques using agar culturing plates with or without overlay of feeding bacteria [36,37]. This makes the test statistically weak for determining AK as a diagnosis. The test, however, is easy to perform and can be cost effective compared to other options. Another aspect to consider is the time that is required to get results back from a culture. Amoebic culture takes, on average, 10 days to demonstrate a positive result [36]. This delay allows clinical morbidities to further progress, making AK more difficult to treat. The procedure can also be invasive if corneal biopsy should be needed to acquire adequate tissue sample for better culture yields.

As an alternative option to culture, corneal scrapes or corneal biopsy can be used in a direct smear with special stains for cytologic examination. Several studies have shown that using microscopy and analyzing a smear with special stains, especially Calcofluor white, has been an effective and rapid method for identification of *Acanthamoeba* cysts [38,39,40]. This method may still require the invasive nature of corneal biopsy. 

### 5.2. Polymerase Chain Reaction

Polymerase chain reaction (PCR) is a molecular technique that is growing as a diagnostic tool for corneal infections including AK. PCR is not used clinically as broadly as corneal culture; however, the diagnostic tool does improve upon statistical strength for diagnosis of AK. PCR demonstrates a high specificity (100%), as well as a moderately better sensitivity than corneal culture, which ranges from 66.7–100% depending on the DNA section used [36,37]. These statistical values trend toward the higher end if multiple PCR tests are run using different DNA segments. PCR also provides faster results, with the average turnover in diagnosis being 5 days [36]. Performing PCR is a moderately technical process and can be more expensive than other diagnostic techniques. Real time PCR is a newer technique that allows for even faster results. While real time PCR carries a similar range of sensitivity and specificity as standard PCR, as well as a high negative predictive value (99.3%), positive predictive values of real time PCR are relatively low (59.1–61.9%) [36]. The technique also requires specialized machinery. Recently, real time PCR is becoming a standard diagnostic procedure mainly due to the delivery of rapid results, a necessity in the diagnosis of AK. However, if time allows, other diagnostic tools are more accurate in diagnosing AK and should be used to confirm results.

### 5.3. In Vivo Confocal Microscopy

Another diagnostic tool recently used in diagnosing AK is in vivo confocal microscopy (IVCM). IVCM is a growing, non-invasive diagnostic tool that is useful for diagnosing conditions of the eye. However, IVCM is not as readily available as some other diagnostic tools. In facilities with IVCM, it is considered to be the first-line method for diagnosis of AK. IVCM has great statistical value in diagnosing AK with a very high specificity (100%) and a high sensitivity, ranging from 85.3–100% [36,41]. This makes the technique very strong at both confirming and excluding AK as a diagnosis. IVCM can also help differentiate other types of keratitis from AK or identify the presence of mixed keratitis. The test also provides rapid results, especially when compared to other diagnostic tools. One limitation to this method is that only *Acanthamoeba* cysts can be recognized using this method, meaning that diagnosis can only be made at later stages of the disease [42]. The test requires proficient imaging expertise and is also not as widely available and may be more expensive than other options. A recent comparison of two of the confocal microscopes that have been used for this analysis, namely the Nidek Confoscan4 (NIDEK technologies, Fremont, CA, USA) and Heidelberg HRT3 Rostock Cornea Module (RCM) (Heidelberg Engineering, Heidelberg, Germany) have revealed a slight advantage to using the Heidelberg RCM [43]. This study also identified several additional limitations with this technique. The most important being the experience of the operator using these instruments to identify AK. Another set of limitations are the small area of the cornea in any particular scan, so it is possible scans were obtained in an area remote from the pathology. Furthermore, stromal inflammation can result in false negatives if the inflammatory cells and edema mask *Acanthamoeba* cysts, or false positives when macrophages are misidentified as *Acanthamoeba* cysts.

### 5.4. Impression Cytology

The least used diagnostic tool for diagnosing AK is impression cytology. The technique is used for dry eye diagnosis by obtaining superficial corneal epithelial cells with nitrocellulose filters and special stains [44]. As shown in Figure 1C, numerous double walled *Acanthamoeba* cysts interspersed can be seen in a sheet of corneal eparchial cells obtained by impression cytology. There is no reported analysis regarding the statistical prowess of impression cytology. While it is relatively non-invasive and highly specific for diagnosing AK, special stains and expertise in cytopathology are required, However, 4–6 h turn-over can be achieved in a well-experienced lab. Another limitation of this technique is that it only surveyed the superficial corneal epithelial tissues and cannot readily detect pathogens in the deeper stroma. It should be acknowledged that recent reports have indicated that cytological analysis is a much more common technique in other countries [45] and has resulted in very high rates of positivity when compared to culturing of amoeba [46].

## 6. Treatment

The *Acanthamoeba* can exist in an actively mobile trophozoite form or a dormant cyst form which is highly resistant to drugs. Biguanides and aromatic diamidines are effective antimicrobial agents for killing the pathogen but must be given together to overcome drug resistance. Neomycin, an antibiotic, is also beneficial but when given alongside other drugs rather than as a standalone [31]. Treatment is usually administered through the form of topical eye drops, given every hour initially for the first few days. The minimum time for treatment is 3–4 weeks, with the dosage of eye drops decreasing to every three hours after the initial few days. Multiple authors recommend continuing treatment for several months to prevent reinfection or recrudescence of the disease [26]. Eliminating the infection is currently a challenge due to the fact that the cyst form of *Acanthamoeba* is resistant to most treatments, and encystment is prone to occur in the more susceptible trophozoites upon antimicrobial therapy [26,31]. It is important to note that one feature of AK often observed is that inflammation worsens initially after starting treatment before improving [28]. The mechanism is unclear, but the increase in inflammation could possibly be caused by the medication toxicity or the yet unidentified antigens released by the dead organisms [28]. There is promising data on medicinal plants suggesting that its use as a nontoxic drug against AK is worth further researching [47,48,49].

### 6.1. Biguanides

Biguanides are useful antimicrobial agents because they can kill both forms of *Acanthamoeba*, trophozoites and cysts. The positively charged molecules bind to and penetrate the amoebas and increase the cytoplasmic membrane permeability resulting in death of the pathogen [26]. The two biguanide compounds consistently proven effective in a drug treatment are polyhexamethlyene biguanide (PHMB) and chlorhexidine. PHMB, a pool disinfectant, at a low concentration of 0.02% has a high cysticidal activity against multiple strains of the pathogen [50]. Chlorhexidine has a slightly lower cysticidal activity than PHMB but may still be a more effective alternative to PHMB because it is a smaller molecule that can invade the stroma more easily [51]. Another advantage of using PHMB and chlorhexidine is that these compounds seem to have less toxicity problems when compared to the aromatic diamidine propamidine [28]. Side effects of this treatment can involve an elevated intraocular pressure and toxic keratopathy [26]. Therefore, patients may have to be monitored for their intraocular pressure and the need for antiglaucoma medication alongside this treatment. 

### 6.2. Aromatic Diamidines

Aromatic diamidines such as propamidine and hexamidine are often used to treat AK in combination with biguanides to prevent drug resistance to diamidines. One of the mechanisms of action of aromatic diamidines probably involves binding to the parasite’s DNA which would result in inhibition of its growth [51]. Propamidine was one of the first treatments discovered for treating AK and has therefore been a part of many different combination therapies. Propamidine has also been successfully administered for AK with antifungal medications such as topical miconazole 1% and oral itraconazole [26,28,52]. Propamidine is found to be the most effective aromatic diamidine, however there are many reports of *Acanthamoeba* developing resistance to propamidine which is why the preferred treatment is to take it alongside a biguanide like chlorhexidine [46,53,54]. 

### 6.3. Antibiotics

Neomycin can eliminate the trophozoite form of Acanthamoeba but does not have a high cysticidal activity like other previously mentioned drugs. This drug cannot be used alone because along with cysts being resistant, neomycin can promote hypersensitivity to itself and cause the development of neomycin-resistant temperature-sensitive mutants [52]. Using neomycin in a “triple therapy” such as with propamidine and dibromopropamidine has effectively treated many AK patients [55,56]. This is because neomycin has an indirect effect on Acanthamoeba by decreasing bacterial food for trophozoites and preventing bacterial superinfection [31].

### 6.4. Steroids

The use of corticosteroids to decrease the intense inflammation that occurs in AK is controversial because of the disadvantage of potentially promoting encystment and in-creasing trophozoites [31]. However, a study reports of finding no evidence of a strong correlation between beginning topical corticosteroids after starting treatment and worsening the outcome of the disease [57]. The topical corticosteroids are used to de-crease pain and improve comfort, however most cases of extracorneal inflammation can be treated with nonsteroidal anti-inflammatory drugs too, such as 50 to 100 mg of flurbi-profen, two to three times daily [31]. Flurbiprofen is an anti-inflammatory drug that can act as an analgesic and mydriatic as well [58]. Corticosteroids suppress the activity of macrophages that could potentially be attacking dead cysts persisting in the corneal stroma causing the intense inflammation in the first place [58]. Although there is no ev-idence against using corticosteroids after beginning treatment, there is data that individuals with a history of corticosteroid use before diagnosis are more likely to fail initial treatment of AK [31].

### 6.5. Surgery

If the recommended treatment of biguanides and aromatic diamidines fail [28,53], and the infection has progressed to an advanced stage, then a therapeutic corneal transplantation or various keratoplasties can be the last resort of treatment. A variation of keratoplasty called Deep Anterior Lamellar Keratoplasty has been suggested as a better surgical option for AK patients for prevention of intraocular invasion by pathogens due to its non-penetrating nature [28]. Corneal cryotherapy and amniotic membrane transplantation are other options if topical treatments fail [31]. Keratoplasty is most useful for patients who are experiencing vision loss due to either the advanced stromal destruction induced by uncontrolled infection or corneal scarring after medical eradication of the infection. 

### 6.6. New Treatment Approaches

While chlorhexidine, PMHB, and propamidine are currently the recommended agents of AK treatment [26,51], a few reports have employed collagen cross-linking using riboflavin and UV lights as a successful adjunct strategy for the conventional topical therapy [29]. It is possible that the photochemical reactions stabilize the collagen and prevent further tissue damage with prevention of pathogen reproduction [29]. 

Plants are also potential therapeutic agents. Many species are advantageous sources of amoebicidal agents because along with having high anti-amoebic activity, they have a much lower toxicity than the drugs currently used to treat AK [47]. There are reports of at least ten different medicinal plants having high trophozoite and cysticidal activity with no toxicity to human keratocytes [47]. Tea tree oil demonstrated 100% effectiveness against both trophozoite and cyst forms of *Acanthamoeba* in vitro [48]. Although further research is required, this semi-in vivo study demonstrates that tea tree oil can potentially destroy amoebae in the cornea through the form of eye drops. Tea tree oil can penetrate tissues therefore, it would be able to attack both shallow and deep layers of the cornea [48]. The oil did not damage the eyes of any of the mice therefore demonstrating no toxicity [48]. An aqueous extract of the nigella sativa plant was also been studied as a potential therapy for AK patients. Nigella sativa has antioxidant activity along with phenolic, alkaloid, and saponin constituents that enhance opsonization and phagocytosis of *Acanthamoeba* [49]. This plant also inhibits bacterial biofilm, and therefore could result in disrupting *Acanthamoeba*’s binding to the cornea [49]. Experiments have proven both of these plants to be non-toxic but studying the effects in a larger sample size is the next step to take for future studies.

Regarding surgical approaches for the resultant corneal scarring, photorefractive keratectomy (PRK) has been reported to result in vision improvement with no disease recurrence in a few patients [26]. 

## 7. Prevention

General prevention paradigm for AK should start with the avoidance of common risk factors. Avoiding contaminated waters and corneal trauma will help minimize risk for the corneal infection by preventing the two most common routes of infection. In those who wear contacts it is imperative to wear and store contacts for the appropriate amount of time. Soft contacts should not be slept in and they should not be stored for more than 12 h. When contacts are to be stored, they should be kept in a contact case filled with hydrogen-peroxide based multipurpose solution and rinsed before being stored [20]. Multipurpose solutions are a potential area of improvement in the prevention of AK if a specific solution can be produced that has higher efficacy against acanthamoeba cysts.

## 8. Summary/Future Direction

*Acanthamoeba* keratitis is a relatively new and under-recognized corneal infection that is common to, but not limited to, contact lens wearers. An earlier diagnosis of this disease is correlated to a better prognosis. As the symptoms are slowly progressing and initially resembling other infections such as herpetic keratitis, there is an urgent need for developing a more reliable laboratory test for AK. This disease can be effectively treated with aromatic diamidines and biguanides in a combination therapy. As there is an increasing prevalence of contact lens usage in the developed world and currently no efficacious monotherapy for AK, further search for newer therapeutic agents or strategies are warranted, However, one needs to be bear in mind that *Acanthamoebas* are phylogenetically similar to humans and therefore make it challenging to find an agent that selectively harms the parasite without harming its host [26]. Taking future steps in augmenting consumer education and public awareness should prevent the occurrence and improve the outcome of AK.

## Figures and Tables

**Figure 1 pathogens-10-00323-f001:**
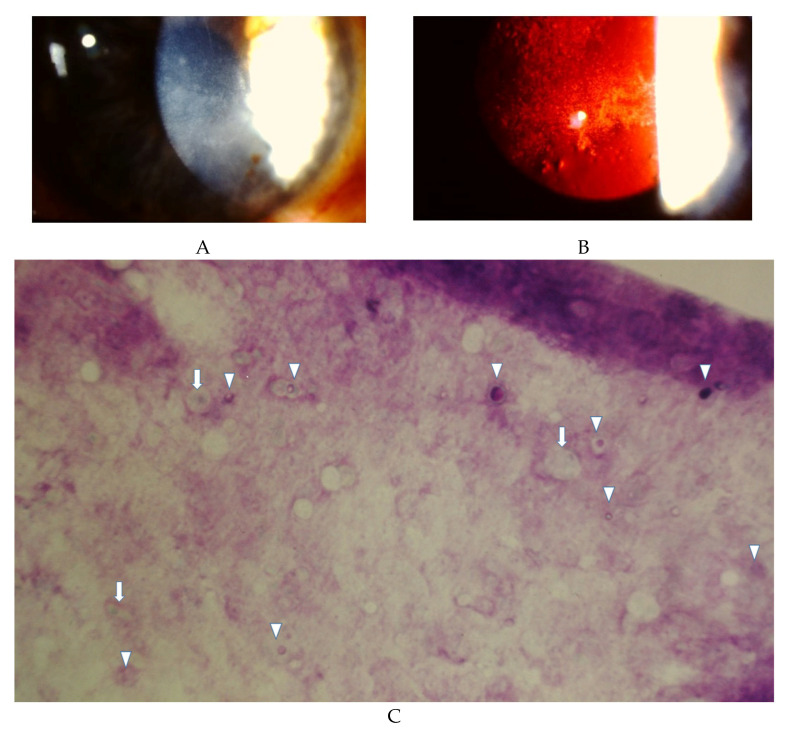
A 40-year-old male contact lens user developed severe ocular irritations after using home-made saline solution for cleansing contact lenses. (**A**) Slit lamp photo showed diffuse, coarse superficial punctate epithelial erosions and mild anterior stromal haze without epithelial defect or stromal infiltrates. (**B**) Slit lamp photo with retro-illumination of the same cornea readily showed the coarse epithelial erosions and relatively clear stroma. (**C**) Impression cytology of the central cornea from the same eye showed multiple double-walled *Acanthamoeba* cysts (triangles) and occasional trophozoites (arrows) scattered among the epithelial cells in the superficial corneal epithelial sheet.

**Figure 2 pathogens-10-00323-f002:**
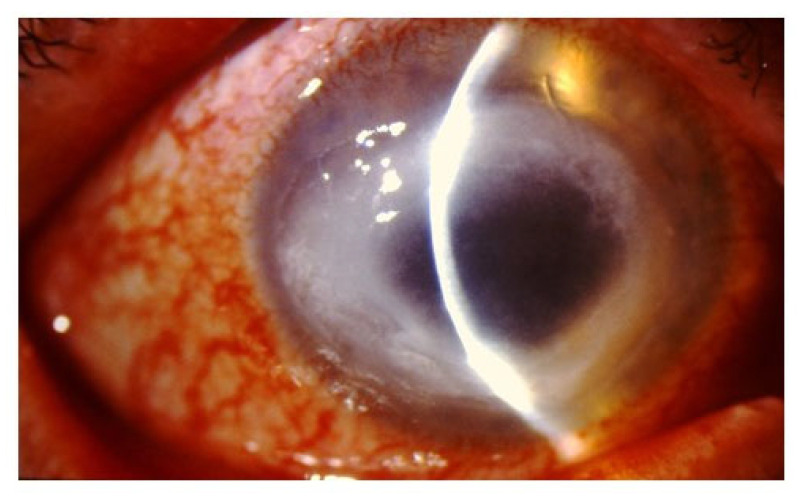
A dense ring corneal infiltrate in a 74-year-old female with a history of extended soft contact lens and intense painful ring corneal ulcer refractory to conventional antibiotic therapy for presumed bacterial corneal ulcer. Her initial corneal culture was negative for microorganisms. Subsequent corneal biopsy revealed multiple *Acanthamoeba* cysts in corneal stroma.

## Data Availability

Not applicable.

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
