# Peer review of "Acanthamoeba Keratitis, Pathology, Diagnosis and Treatment"

_pathogens, 2021, doi:10.3390/pathogens10030323_

Round 1

Reviewer 1 Report

The authors have satisfactorily responded to my critique.

Author Response

Dear Pathogens editors,

Thank you once again to the editors and reviewers for the recommended minor edits to the manuscript #1132922. We have adjusted it, accordingly, please see the specific comments down below.

1.) Line 199 - 200.  Real time PCR - you state that the statistical values are not as strong but give no figures despite recommending it in line 200- 201.

Statistical values have been added for real time PCR and the position on the technique has been clarified, please note lines 201-207.

2.) Line 257 - should give references.

The specific references have been added, note line 263.

3.) neomycin misspelt in lines 287 and 288

This spelling error has been adjusted, note lines 293-294.

4.) The present manuscript is an extensive review related to Acanthamoeba keratitis. The authors should include the importance of Acanthamoeba genotypes in their review.

The significance of Acanthamoeba genotypes has been added to the manuscript, please note lines 44-50.

Once again, on behalf of my fellow authors, I wish to thank you for your consideration of our manuscript.

Sincerely,

Patrick Stuart, Ph.D.

Saint Louis University

Reviewer 2 Report

The authors have made a number of changes to improve the MS

there are a couple of minor issues.

  1. Line 199 - 200.  Real time PCR - you state that the statistical values are not as strong but give no figures despite recommending it in line 200- 2001

2. Line 257    - should give references

3. neomycin misspelt in lines 287 and 2888

Author Response

Dear Pathogens editors,

Thank you once again to the editors and reviewers for the recommended minor edits to the manuscript #1132922. We have adjusted it, accordingly, please see the specific comments down below.

1.) Line 199 - 200.  Real time PCR - you state that the statistical values are not as strong but give no figures despite recommending it in line 200- 201.

Statistical values have been added for real time PCR and the position on the technique has been clarified, please note lines 201-207.

2.) Line 257 - should give references.

The specific references have been added, note line 263.

3.) neomycin misspelt in lines 287 and 288

This spelling error has been adjusted, note lines 293-294.

4.) The present manuscript is an extensive review related to Acanthamoeba keratitis. The authors should include the importance of Acanthamoeba genotypes in their review.

The significance of Acanthamoeba genotypes has been added to the manuscript, please note lines 44-50.

Once again, on behalf of my fellow authors, I wish to thank you for your consideration of our manuscript.

Sincerely,

Patrick Stuart, Ph.D.

Saint Louis University

This manuscript is a resubmission of an earlier submission. The following is a list of the peer review reports and author responses from that submission.

Round 1

Reviewer 1 Report

This review is well-written for the most part, but the content and depth of the discussions are unsatisfactory.  There are essentially no new insights and this paper is a re-hash of previous review papers on this subject. Unfortunately, some topics of clinical relevance are completely ignored.  For example, the dilemma as to when or whether to utilize corticosteroids to dampen the intense inflammation that occurs in Acanthamoeba keratitis is not mentioned even though this is a highly relevant issue among clinicians.   Excellent retrospective studies on the effect of corticosteroids in Acanthamoeba keratitis have analyzed the timing and effects of topical steroids and anti-microbials (Carnt et al. Ophthalmology  2016; Robaei et al. Ophthalmology 2014).

The images offer no new insights and similar images of superior quality can be found in any number of publications on Ophthalmology and in a wide range of textbooks. Figure 1C is unimpressive and the legend states that trophozoites and cysts are present. The image is blurred and one would have to use a great deal of imagination to identify cysts or trophozoites in this image. No arrows are included to identify the cysts and trophozoites.

Minor points

  1. The genus name Acanthamoeba should be italicized.
  2. The first sentence in the abstract stating that “Acanthamoeba keratitis is a relatively recent and unusual corneal infection..” is misleading. Acanthamoeba keratitis was described over 45 years ago (Jones DB, Visvesvara GS and Robinson NM. Trans Ophthalmol Soc UK UK 95:221,1975).

Reviewer 2 Report

This review needs some grammatical editing.

Line 61. Full stop before However.

Lines 62-64 – unclear what this sentence means – recommend multipurpose contact lens solutions but these have limited efficacy.  What should be used then in terms of solutions eg hydrogen peroxide containing cleaners ?

The absence of a pathogenesis section means that important information regarding the etiology and course of the infection is not given. There is no discussion of the two forms, trophozoites and cysts, and change between the two forms and different response to therapy.

Signs and symptoms – this does not mention the “dirty epithelium” like changes or explain the differences between the pseudodendritic epitheliopathy from true viral dendrites which have the rounded spot like widenings of the endings of the epithelial erosions.

Line 91  signs of AK distinct from rather than unique from

The section on diagnostics is incomplete. The authors only mention corneal scrapings in the context of culture which takes at least 10 days but omit the use of direct smears with special stains such as Calcofluor white and histopathology of biopsy with special stains. The authors do not mention that in about 23% of cases, a mixed infection with virus, bacteria and fungi is seen and this is important for management.

Lines 122- 123 is weak.  

PCR is becoming more standard procedure for diagnostics and real time PCR in particular.

In vivco corneal Tandem scanning confocal microscopy is now considered firstline method of diagnosis where facilities exist.

Treatment.  The authors do mention use of neomycin but do not mention triple therapy with biguanides, aromatic diamidines and neomycin which is used in Germany.  Neomycin is useful not only in killing trophozoites but reduction of the bacterial load which is  a food source for Acanthamoeba and preventing bacterial superinfection. The authors do not give an indication of how long therapy should be maintained for – just a minimum of 3-4 weeks whereas other reviews suggest a much longer time of 6 – 12 months.

Surgical therapy could also include corneal cryotherapy as adjuvant treatment of topical therapy.

In summary, compared with other recent reviews on this topic, this article is deficient in a number of areas (pathogenesis, prevention, diagnosis and Treatment)  and would not be the reviewer’s first choice of recommended literature for information on this important topic

Reviewer 3 Report

The author presents a literature review to describe the current diagnosis and management of Acanthamoeba keratitis. The manuscript is well written, however, some text is repetitive.  Paper has an important clinical message and should be of great interest to the readers although, in the reviewer opinion from the medical field.

Before publication, a few considerations should be taken into account.

Comments to authors

General comments:

The manuscript is in general well-organized but could be tightened up a bit to make more evident and transparent what the new contribution is.

Section introduction:

I believe that some re-writing needs to be done in the introduction section. The opening paragraphs are too general. First, concerning the introduction, this reviewer feels that too little information is provided with regard to Epidemiology  (with reported rates).

Acanthamoeba keratitis (AK) is a potentially sight-threatening infection of the ocular surface that is produced by several amebas of the genus Acanthamoeba; 8 species, 5 genotypic classes have been reported to cause keratitis, this information is missing in the reviewer's opinion.

Section risk factors

The section risk factors is well written, however, some text is repetitive.

 Section sings/symptomps

Fig. 1 C: Impression cytology of the central cornea from the same eye showed multiple double-walled Acanthoamoeba cysts and occasional  trophozoites scattered among the epithelial cells in the superficial corneal epithelial sheet, could be an exchange, in my opinion (scale/ staining?);

And please change Acanthoamoeba versus Acanthamoeba 

Section diagnosis and treatment

AK remains a challenging condition to diagnose and treat. in the opinion of the reviewer, there was no indication of the areas for further research include genetic therapy, new markers, new targets for drug therapy, and stem cell research.